# Permselectivity of Cation Exchange Membranes Modified by Polyaniline

**DOI:** 10.3390/membranes11030227

**Published:** 2021-03-23

**Authors:** Irina Falina, Natalia Loza, Sergey Loza, Ekaterina Titskaya, Nazar Romanyuk

**Affiliations:** Physical Chemistry Department, Faculty of Chemistry and High Technologies, Kuban State University, 350040 Krasnodar, Russia; nata_loza@mail.ru (N.L.); s_loza@mail.ru (S.L.); katezolotka@mail.ru (E.T.); romanyuknazar@mail.ru (N.R.)

**Keywords:** cation exchange membrane, polyaniline, permselectivity, electrodialysis, characterization

## Abstract

This work discusses the applicability of polyaniline-modified cation exchange membranes for the separation of monovalent/divalent cations by electrodialysis. A novel method of membrane modification directly in the electrodialysis unit is used to prepare permselective membranes. Complex characterization of the membranes before and after modification allows revealing the influence of membrane matrix on the modification efficiency. The characterization of the membranes includes determination of the diffusion permeability, specific conductivity and current–voltage curves in HCl, NaCl and CaCl_2_ solutions, as well as transport-structural parameters of the extended three-wire model. The characterization results are used to predict the influence of the modification on membrane permselectivity. The competitive mass transfer of singly and doubly charged cations in the electrodialysis process is investigated in underlimiting and overlimiting currents. Electrodialysis desalination of a solution containing Na^+^/Ca^2+^ or H^+^/Ca^2+^ cations shows that the modification leads to an increase in membrane permselectivity to single-charged cations due to the repulsion of Ca^2+^ ions from the positively charged membrane surface. The permselectivity of the polyaniline-modified perfluorinated membrane to H^+^ in the mixture of H^+^/Ca^2+^ cations is observed in all current regimes.

## 1. Introduction

Progress poses new challenges for industry and science, associated with a decrease in the ecological load on the environment and depleting the natural resources. The electromembrane technologies could respond to such challenges due to the possibility of engineer the closed water circulating systems, preparing the water with the required grade of purity, and separating mixed solutions [1,2,3,4]. The separation of singly and multiply charged ions is one of the major objectives due to the opportunity to recover the valuable components and return them to industrial processes. The cation exchange membrane (CEM) selectivity to singly charged cations in a mixture of monovalent/divalent cations could be effectively improved by introducing the positively charged anion exchange groups into its structure. Multiply charged cations are more intensively repulsed by anion exchange groups in comparison with singly charged ones, and CEM selectivity to monovalent cations (permselectivity) increases. The first type of such materials is bifunctional zwitterion CEM. It contains a small fraction of chaotically distributed anion exchange groups, which are introduced during the membrane synthesis [5]. The other type is the membrane, which surface is covered by a thin, positively charged polyelectrolyte layer using layer-by-layer deposition [6]. This method enables modifying the commercial electrodialysis membranes and is commercially feasible in comparison to direct membrane synthesis. The promising method of improving the membrane permselectivity of membranes is the synthesis of polymer modifiers on the surface of the monopolar membrane. Conjugated polymers are often used as a modifier for this purpose. They are applied on the surface of CEM by template polymerization due to the simplicity of the synthesis and durable immobilization of the modifier on the membrane surface as interpolymer complex “polyelectrolyte—conjugated polymer” [7,8,9,10]. In addition, in some cases, the modification could be performed directly in the electrodialysis unit [11].

A quite efficient modifying agent for this aim is polyaniline (PANI) due to its high conductivity, chemical and thermal stability [12]. The efficiency of the application of the PANI-modified ion exchange membrane in the separation of singly- and multiply-charged ions (including acid solutions) has been shown in [13,14,15,16,17,18,19]. The size of the membrane operating area in those papers is insufficient for scaling the experimental results to industrial electrodialysis apparatus.

The main role in the properties of the PANI-modified ion exchange membranes plays the content and distribution of modifiers in the membrane phase. The properties of composites depend on the PANI synthesis conditions: concentrations of aniline and oxidant, synthesis duration, the nature of oxidant, background acid and solvent, presence of concentration and electric fields, etc. [20,21,22,23,24,25]. The PANI could localize on the surface or in the bulk of the membrane according to the type of oxidizer: co-ion or counter-ion towards the initial membrane [26]. Thus, the variation of the conditions of the PANI synthesis enables obtaining composites with the different content and distribution of the modifier.

The regularities of composites formation are mainly investigated for Nafion-type membranes due to the variety of their applications, such as membrane electrolysis, proton exchange membrane fuel cells, sensors and others [27,28,29,30]. The application of such membranes in electrodialysis is unusual due to their high cost. However, the application of perfluorinated membranes in the electrodialysis is justified for the treatment of aggressive media, where the commercial electrodialysis membranes are unstable.

It is known that the ionic form of the membrane affects its transport properties [31,32], but transport and structural characteristics of the PANI-modified membranes in solutions containing multiply charged counter-ions have not been still investigated. In the case of multicomponent solutions, the competitive transfer of the different counter-ions should be taken into account. The membrane characterization opens the possibility to predict the regularities of competitive mass-transfer of the different counter-ions in the electrodialysis on the basis of ordinary experimental methods, such as determination of diffusion permeability (DP), specific conductivity, current–voltage curves (CVCs) and others. The estimation of parameters of the extended three-wire model on the basis of conductivity concentration dependencies enables evaluating the counter-ions transport numbers and complement the characterization. These problems, however, have not been discussed in the literature. This work is aimed at the complex characterization of composite CEMs modified by the PANI in solutions of singly/multiply charged cations and to the estimation of the efficiency of modified membrane application for cations separation in the electrodialysis.

## 2. Materials and Methods

### 2.1. Objects of Research

The objects of research were homogeneous perfluorinated MF-4SK (JSC Plastpolymer, Russia) and heterogeneous polystyrene MK-40 (LLC “Innovative Enterprise Shchekinoazot”, Tula Region, Russia) membranes. Both membranes contained sulfonic acid groups to eliminate the influence of fixed ions on modification efficiency and transport properties. The heterogeneous polystyrene MA-41 membranes (LLC “Innovative Enterprise Shchekinoazot”, Tula Region, Russia) containing quaternary amino-groups served as auxiliary membranes in the electrodialysis demineralization process.

The MK-40 membrane consists of KU-2 ion exchange resin based on divinylbenzene crosslinked polystyrene and polyethylene as the inert binder. This membrane is designed for the solution treatment by electrodialysis [33,34]. The membrane is reinforced by the caprone fabric to impart mechanical strength. The MF-4SK membrane is the perfluorinated homogeneous-type membrane. Perfluorinated membranes are stable in acid and aggressive media, unlike polystyrene ones. It has complex polymer architecture and consists of hydrophobic polymer chains and uniformly distributed sidechains with highly hydrophilic sulfonic acid groups [35]. The main physicochemical characteristics (ion exchange capacity (IEC), water content, density, specific water content) of the MK-40 and MF-4SK membranes in the swollen state are presented in Table 1. The used experimental techniques of determining the IEC, water content and specific water content are described elsewhere [36]. The membrane density was measured by the hydrostatic weighing method.

The conditioning of homogeneous membranes was performed by the thermal-oxidative method [36] and included the successive boiling of the membrane in 5% HNO_3_ solution, 10% H_2_O_2_ solution and distilled water, the duration of each stage was 3 h. The membrane was washed with distilled water between stages. The pretreatment of the heterogeneous membranes comprised the degreasing of the membrane surface by carbon tetrachloride and its successive immersion in ethanol, sodium chloride solutions with concentrations 300, 100 and 30 g/L and distilled water for 24 h in each step. Afterward, the membrane was placed in a 10% solution of HCl for 48 h and washed with distilled water.

### 2.2. Modification of Membranes by the PANI

CEMs were modified by the PANI directly in the electrodialysis unit [37]. The cell consisted of 7 cation and 8 anion exchange membranes. Figure 1 shows the synthesis scheme. The 0.01 M aniline solution in 0.05 M sulfuric acid circulated in the desalination chambers. The 0.008 M ammonium persulfate solution in 0.025 M sulfuric acid circulated in the concentration chambers. The feeding tanks for concentration and desalination chambers contained 10 L of solutions. The synthesis was carried out in two steps. First, the system was exposed under 2 A/dm^2^ for 10 min, then for 1 A/dm^2^ for 120 min for the MK-40 membrane and 50 min for the MF-4SK one. Then membranes were washed with sulfuric acid solution and water. The composites, prepared under the current conditions, contain PANI in emeraldine salt form [26]. Before experiments, modified samples were placed in acid solution to remove residual aniline and washed with distilled water. Table 1 demonstrates the physical-chemical characteristics of the initial and modified membrane samples.

### 2.3. Diffusion Permeability

The DP was measured in two chamber cells separated by the membrane under investigation (the membrane area was 21 cm^2^) [38]. One chamber was filled with the solution having a certain concentration, and the other—with water. The solutions were stirred during the experiment to eliminate the influence of diffusion layers on electrolyte transport in the solution/membrane interface. The electrolyte diffusion process was controlled by measuring the resistance in the chamber containing water by an immittance meter (RLC) E7-21 (MNIPI, Minsk, Republic of Belarus). The DP coefficient (*P*) was calculated according to the formula:(1)$$P=\frac{V_{w}\cdot l}{S\cdot C_{0}}\cdot \frac{dC}{dt},$$
where *V*_w_ is the volume of the chamber with water; *C* and *C*_0_ are current and initial concentrations of the solution, correspondingly; *l* and *S* are thickness and working area of the membrane. Using dilute solutions enables determining the rate of an increase in the solution concentration in the chamber water containing (*dC*/*dt*) from the slope of the kinetic dependency of the solution conductivity in this chamber tgα=d(1R)dt by the formula:(2)dCdt=d(1R)dt⋅K,
where *K* is the cell constant corresponding to the slope of the concentration—conductivity dependence *C*—1/*R*.

### 2.4. Specific Conductivity

The specific conductivity (κ_m_) was calculated from the active part of the membrane resistance under alternative current measured by mercury contact method [38] using the potentiostat-galvanostat Autolab PGSTAT302N (Metrohm Autolab B.V., Utrecht, The Netherlands) equipped by FRA-32 impedance unit by the formula
(3)$$\kappa_{m}=\frac{l}{R\cdot S},$$
where *R* is the membrane resistance, the membrane working area *S* was 0.785 cm^2^. The samples were equilibrated with the studied solutions before measurements.

### 2.5. Current–Voltage Curves

The membrane behavior under conditions close to reality can be estimated by the voltammetry method. The CVCs of the initial and modified membranes were measured in solutions of HCl, CaCl_2_, NaCl and their equivalent mixture. The total solution concentration was 0.05 mol-eq/L in all cases. The CVCs of membranes were measured in the galvanodynamic regime at a scanning rate of 10^−4^ A/s. Direct current was applied to the platinum polarizing electrodes using the Keithley 2420 SourceMeter (Keithley Instruments, Inc., Cleveland, OH, USA). The potential drop across the membrane understudy was measured using Ag/AgCl electrodes by the Keithley 2701 ethernet multimeter/data acquisition system (Keithley Instruments, Inc., Cleveland, OH, USA). These electrodes were placed in the Luggin–Haber capillaries. The Luggin–Haber capillaries were installed at both sides of the membrane under study in its geometric center at a distance of about 0.5 mm from the surface. The measurements of CVCs for ion exchange membranes occur under conditions of the laminar solution flow with volume crossflow velocity equal to 14 mL/min. All the experiments were performed at least three times to average the results. The experimental error did not exceed 5%.

### 2.6. Electrodialysis Treatment of Mixed Solutions

The investigation of the electrodialysis demineralization process was performed in a laboratory electrodialysis unit equipped with hydraulic manifolds for concentration and desalination chambers. The characteristics of the electrodialysis unit were determined according to the method [39], which allows scaling the results for laboratory unit to industrial one by compartmentation method. The unit included five elementary cells consisting of the cation- and anion exchange membranes each. The working area of each membrane was 0.01 m^2^. CEMs were commercial MK-40 and MF-4SK membranes and composites on their base with the PANI (MK-40/PANI and MF-4SK/PANI); the anion exchange membranes were commercial MA-41 membranes. The modified membranes faced the counter-ions flux by the PANI layer. The distance between membranes in the electrodialysis unit was 0.9 mm, and the membranes were separated by the grids; the solution linear velocity was 0.02 m/s. The operating regime of the laboratory electrodialysis unit was identical to the working conditions of industrial electrodialysis units. The electrodialysis process was performed in a potentiostatic regime under 0.75, 1, 1.5, 2, 4 and 6 V per elementary cell. The concentrations of Na^+^ and Ca^2+^-ions at the inlet and outlet of the concentration and desalination tracts were controlled every 60 min by a Stayer ion chromatographer (JSC Akvilon, Moscow, Russia), H^+^-ions concentration was determined by pH-titration using the automated titrator EasyPlus titrator Easy Pro (Mettler-Toledo International, Inc). The kinetic dependencies of the cation concentration during the electrodialysis of the mixed solutions (Na^+^ and Ca^2+^; H^+^ and Ca^2+^) were used to calculate the main mass transfer characteristics (ion fluxes *J_i_*, current efficiency *η* and specific energy consumption *W*) as follows:(4)$$J_{i}=\frac{\Delta C_{i}\cdot V}{t\cdot S\cdot N},$$
(5)$$\eta =\frac{F\cdot V\cdot \Delta C_{i}}{n\cdot \int_{0}^{t}Idt},$$
(6)$$W=\frac{U\cdot \int_{0}^{t}Idt}{q\cdot t},$$
where *J_i_* is integral ion flux of *i* ions; Δ*C_i_* is changes of *i* ions concentration during the experiment; *V* is the volume of the desalination chamber; *N* is number of CEMs in the electrodialysis unit; *F* is Faraday constant; *I* is current intensity; *n* is a number of elementary cells in the electrodialysis unit; *η* is current efficiency; *W* is energy consumption for the treatment of 1 m^3^ of the solution; *U* is the voltage applied on the electrodialysis unit; *q* is solution volume velocity.

## 3. Results and Discussion

### 3.1. Influence of the PANI on Transport Properties of Ion Exchange Membranes

To reveal the influence of polyaniline surface layer on transport properties of CEMs with the different structures, the complex characterization of the membranes in NaCl, HCl and CaCl_2_ solutions was performed. The characterization includes investigation of the concentration dependencies of the DP and conductivity, CVCs, SEM images of the membrane surface, as well as the estimation of transport-structural parameters and counter-ion transport number within an extended three-wire model.

#### 3.1.1. Diffusion Permeability

Figure 2 and Figure 3 present the concentration dependencies of the DP in HCl, NaCl and CaCl_2_ solutions for the initial and modified membranes. The DP values for the initial homogeneous MF-4SK membrane are higher than for the heterogeneous MK-40 one in all electrolyte solutions. Diffusion experiments for modified membranes are performed at the different orientations of the membrane towards the electrolyte flux (by modified and non-modified surfaces) due to their anisotropic structure. The asymmetry of diffusion characteristics for the MF-4SK/PANI membrane is not detected (Figure 2). Two concentration dependencies are presented for the MK-40/PANI composite membrane in each solution (Figure 3b). DP of charged polymers depends on the diffusivity of co-ions. In the case of CEMs, the diffusion characteristics of the membrane depend on the diffusivity of anions. In our experiment, the electrolytes with similar anions are used, so the DP value is determined by the nature of the counter-ions.

Figure 2 demonstrates that the DP for homogeneous MF-4SK membrane decreases after modification 2–3 times in HCl, in order of the magnitude in NaCl and in two orders of the magnitude in CaCl_2_ solutions. In the case of a modification of the heterogeneous MK-40 membrane, the effect of diffusion reduction is less pronounced (Figure 3). The DP dependencies in CaCl_2_ solution decrease with concentration growth for the MK-40 and MK-40/PANI membranes, unlike in NaCl and HCl solutions. As can be seen from Figure 3, the DP decreases ≈1.5 times in comparison with the initial membrane when the non-modified surface of the MK-40/PANI membrane faces the electrolyte flux and in 2–3 times for the inverse orientation. Accounting that the electrolytes have similar anions and equal equivalent concentrations, the diffusion asymmetry order of cations is Ca^2+^ < H^+^ < Na^+^. The main reason for the DP decrease is the formation of the positively charged barrier PANI layer on the membrane surface, so the basic matrix retards the anions, and the PANI layer slows the cation transfer. From the diffusion data, one can assume that the most effective separation of singly and doubly charged cations could become apparent in the case of a modified MF-4SK membrane in the mixed HCl + CaCl_2_ solution.

#### 3.1.2. Specific Conductivity

Membrane conductivity is one of the most important membrane characteristics because it may influence the resistance of the electromembrane system and energy consumption during electrodialysis. The conductivity concentration dependencies for the initial and PANI-modified membranes are presented in Figure 4. The essential decrease in specific conductivity (in 3–4 times of the magnitude) is observed after modification of the homogeneous membrane by the PANI in all solutions. At the same time, the conductivity of the heterogeneous MK-40 membrane reduces 1.4 times.

#### 3.1.3. Extended Tree-Wire Model

The membrane structure could be presented as a micro-heterogeneous two-phase system [40]. The first conducting phase includes the swollen polymer, i.e., polymer chains, fixed groups and bound water, and has the unipolar conductivity. The second conducting phase is the internal “free” equilibrium solution with bipolar conductivity similar to the external solution. The phases do not have a definite interface boundary and present themselves as pseudo-phases with chaotic mutual spatial orientation. According to the extended tree-wire model, the current paths in the ion-exchange membrane through three parallel channels: gel phase, intergel solution and mixed solution with the interchange of gel and solution. The conductivity of such a system could be described as follows:(7)$$K_{m}=\frac{aK_{d}}{e+dK_{d}}+bK_{d}+c,$$
(8)$$b=f_{1}{}^{1/\alpha },$$
(9)$$c=f_{2}{}^{1/\alpha },$$
(10)$$a=1-f_{2}{{}^{1/}}^{\alpha }-f_{1}{}^{1/\alpha },$$
(11)$$d=1-(f_{1}-b)/a,$$
(12)$$e=\left(f_{1}-b\right)/a,$$
where *K_m_*, *K_d_* are dimensionless conductivities of the membrane and its gel phase, correspondingly, $K_{m}=\frac{\kappa_{m}}{\kappa_{\mathrm{sol}}}$, $K_{d}=\frac{\kappa_{\mathrm{iso}}}{\kappa_{\mathrm{sol}}}$; κ_iso_, κ_sol_ are conductivities of membrane gel phase and equilibrium solution, correspondingly; *f*_1_ and *f*_2_ are volume fractions of gel and intergel phases, correspondingly, *f*_1_ + *f*_2_ = 1; α is a structural parameter characterizing the spatial orientation of the phases inside the membrane (α could possess the value in the range from −1 to 1 corresponding to serial and parallel orientation of conducting phases); *a*, *b* and *c* are fractions of the current passing through the mixed, gel and solution channels, *a* + *b* + *c* = 1; *d* and *e* are fractions of the solution and gel in the mixed channel, *d* + *e* = 1. Considering the gel phase ideal selective, i.e., counter-ion transport number in gel phase equals 1, the counter-ion transport number in the membrane (${\bar{t}}_{+}$) could be determined as follows:(13)$${\bar{t}}_{+}=1-t_{-}\frac{c}{K_{m}},$$
where *t*_−_ is the co-ion transport number in the solution. This model is thoroughly described in [41,42] and enables estimating the set of transport-structural parameters of ion exchange membranes from concentration dependency of its conductivity.

The conductivity concentration dependencies are used to estimate the transport-structural parameters of the extended tree-wire model by Equations (7)–(12), parameter values are presented in Table 2. The table shows that intergel volume fraction (*f*_2_ parameter) has a higher value for the heterogeneous membrane than for the homogeneous one in all electrolyte solutions. It is concerned with the specific preparation technologies: the homogeneous membrane is the continuous selective conducting phase, while the heterogeneous membrane contains grains of ion exchange resin and polyethylene and free solution located at the junction between these polymers. Therefore, the value of the current transport through the gel phase is greater for homogeneous membranes. Table 2 shows that the volume fraction of the intergel phase is almost independent of the ionic form of the membrane, and the value of the α parameter fluctuates within the range of 0.3–0.6 due to the differences in the sample structure.

The modification of the membranes by the PANI leads to an increase in values of the intergel volume fraction for both types of membranes. This fact affects the current transport through the mixed channel (*a* parameter) in NaCl and CaCl_2_ solutions and portion of the solution in the mixed channel (*d* parameter) in acid solution. The value of the *c* parameter increases for all electrolyte solutions. The effect of modifier on transport properties and model parameters of the initial membrane is more pronounced in the case of the homogeneous membrane since it consists of flexible polymer chains, while the heterogeneous membrane has a rigid structure. According to the model, a modifier is incorporated into the gel phase. Accounting for the data presented in Table 2, one can conclude that the PANI displaces water from the gel phase of the membrane due to its electrostatic interaction with the fixed negatively charged groups of the initial membrane.

The quantitative characteristic of ion exchange membrane selectivity in the solution of individual electrolytes is the true transport number of counter-ions. Value of *c* parameter enables calculating the concentration dependencies of true transport numbers of counter-ions by Formula (13). Calculation results are presented in Figure 5. It is known that the perfluorinated homogeneous membrane has higher selectivity than the heterogeneous one. Based on the values of counter-ion transport numbers, we can form the order of the cation selectivity: H^+^ > Na^+^ > Ca^2+^. This order correlates with the increase in radii in hydrated form and hydration numbers of ions.

The modification leads to the decrease in the selectivity of the membranes, which correlates with the rise of the *c* value. Positively charged emeraldine salt has an ion exchange function, so its presence in CEM leads to a decrease in the transport number of cations. Under the current preparation conditions, PANI content in the membrane is about 1 wt %, so the membrane retains cation exchange properties. The selectivity decrease is more pronounced in CaCl_2_ solution in comparison with HCl and NaCl solutions. It should be noted that obtained membranes have an anisotropic structure, while conductivity measurements were performed under alternative current. Retarding effect towards Ca^2+^-cations could be more pronounced under direct current when the PANI layer faces the cations flux.

#### 3.1.4. Current–Voltage Curves

The CVCs of the initial membranes are typical and have three regions [43]: ohmic region, plateau and overlimiting current region (Figure 6). The main characteristic of CVC is the limiting current density, which determines the effectiveness of electromembrane processes. As is expected, the values of the limiting current density (*i*_lim_) are higher in the solution containing HCl than in other solutions because of the abnormal mobility of hydrogen cations. On the whole, the *i*_lim_ values in all solutions for the MF-4SK membrane are slightly higher than for the MK-40 one.

It is known that the presence of the PANI layer on the membrane surface leads to a change in the shape of CVC [44]. If a greater PANI quantity is located on the membrane surface, a more significant asymmetry of CVC is observed [24,25]. The typical changes of the CVCs of the MF-4SK/PANI membrane are found (Figure 7a): the asymmetry of CVCs, the decrease in the magnitude of the *i*_lim_ values on the modified membrane surface, and the absence of the *i*_lim_ on the non-modified surface (Figure 7a). All effects are manifest in hydrochloric acid solution. The reason for these effects is the formation of an internal bipolar boundary layer between PANI and MF-4SK. However, in the case of the MK-40 membrane, these changes are not observed (Figure 7b), and the CVCs have a typical shape in investigated solutions. The *i*_lim_ values of the MK-40/PANI membrane are slightly lower than the initial one, but this decline is negligible. Probably, the amount of the PANI on the MK-40 surface is insufficient to change the CVC shape.

It can be assumed that only MF-4SK/PANI composites will be effective for separating singly and multiply charged ions.

#### 3.1.5. Scanning Electron Microscopy

SEM images of the modified and non-modified surfaces of the MK-40/PANI and MF-4SK/PANI membranes are presented in Figure 8. One can see that the modified surface contains PANI inclusions, while the non-modified surface has morphology identical to the initial membrane. The difference in the morphology of the surfaces confirms the anisotropic structure of composite membranes. The first step of membrane modification is its saturation with anilinium ions, so only ion exchange regions are modified. Due to the homogeneous structure of the perfluorinated MF-4SK membrane, the uniform layer of the PANI is formed on its surface. At the same time, the PANI layer on the surface of the MK-40 membrane is spotted. Heterogeneous MK-40 membrane consists of ion exchange resin and polyethylene (inert binder), so PANI is formed in the region of the ion exchange resin. These results agree with [45].

### 3.2. Electrodialysis Experiments

#### 3.2.1. Heterogeneous Membranes

Heterogeneous membranes are specially designed for demineralization and concentration of electrolyte solutions by electrodialysis. Therefore, the electrodialysis demineralization of the mixed solution contained single-/double-charged cations is investigated. The main characteristics of electrodialysis desalination are current efficiency, energy consumption and permeability coefficient for the separation of the different counter-ions. Figure 9 shows the energy consumption for the electrodialysis of the mixed solutions using the initial and modified MK-40 membrane. As can be seen, despite the decrease in membrane conductivity after modification, shown in Figure 4a, the energy consumptions for the electrodialysis process with the MK-40 and MK-40/PANI membranes are similar.

Figure 10 presents the partial for each cation and total dependencies of the current efficiency of the electrodialysis with the initial and modified MK-40 membranes as the function of potential applied on the elementary cell (*U*_el.c._). As it is expected, the current efficiency for Ca^2+^-ions exceeds the one for Na^+^ in the case of the initial MK-40 membrane at low potential per elementary cell (0.75–1.5 V). When the potential per elementary cell exceeds 2 V, the current fraction transferred by Na^+^-ions increases and the current efficiency for both cations is close. Similar results were observed for the MK-40/PANI membrane in NaCl/CaCl_2_ solutions, but in the low potential range, the difference in the current efficiency for Na^+^ and Ca^2+^ is smaller than for the MK-40 membrane. This effect can be attributed to the repulsion of doubly charged cations by the positively charged PANI layer on the surface of CEM. The total current efficiency for the initial and composite membranes is high and does not depend on the potential per elementary cell.

The other situation was observed in the solution containing a mixture of HCl and CaCl_2_. The current is mainly transported by H^+^ ions in the investigated potential range due to the relay-race mechanism of its transfer in water solution (Figure 10c,d). There is a slight decrease in total current efficiency in high-intensity current regimes in comparison with the electrodialysis of the mixed solution of salts (Figure 10a,b). This effect is observed for both initial and modified membrane and caused by the decrease in membrane selectivity at high potentials per elementary cell.

The main purpose of the work was to evaluate the possibility of the application of the PANI-modified CEMs for the separation of cations in the electrodialysis. The theoretical ratio of fluxes of doubly and singly charged cations in limiting conditions for ideally selective membrane (limiting permeability coefficient $P_{1/2}^{\lim}$) could be estimated on the basis of diffusion coefficients of the cations in the solution according to the formula [46]:(14)$$P_{1/2}^{\lim}=\frac{J_{1}}{J_{2}}=\frac{(z_{1}-z_{A})\cdot D_{1}}{(z_{2}-z_{A})\cdot D_{2}},$$
where *J*_1_ and *J*_2_ are fluxes of 1 and 2 counter-ions, correspondingly; *z*_1_ and *z*_2_ are charges of 1 and 2 counter-ions, correspondingly; *z*_A_ is a charge of co-ion; *D*_1_ and *D*_2_ are diffusion coefficients of 1 and 2 counter-ion in the solution, correspondingly. The diffusion coefficients for H^+^, Na^+^ and Ca^2+^ ions in the solution are 9.34 × 10^−9^, 1.34 × 10^−9^ and 0.79 × 10^−9^ m^2^/s [47]. $P_{Ca^{2+}/H^{+}}^{\lim}$ and $P_{Ca^{2+}/Na^{+}}^{\lim}$ values calculated according to Equation (14) equal to 0.13 and 0.89 correspondingly. The experimental permeability coefficient (*P*_1/2_) could be estimated as the ratio of fluxes of the different counter-ions divided on their equivalent concentrations in solutions fed to the inlet of the desalination tract.

Figure 11 shows the flux ratio (permeability coefficient) obtained from the electrodialysis experiments for the initial and modified membranes. The results of the electrodialysis desalination of the mixed NaCl + CaCl_2_ solution using initial membranes have shown that in the low potential range, Ca^2+^ flux significantly exceeds the Na^+^ flux. For example, the Ca^2+^/Na^+^ flux ratio is 2–3 for both types of membranes at 0.75 V per elementary cell due to the higher mobility of doubly charged ions in the solution. At higher potentials per elementary cell, the Ca^2+^/Na^+^ flux ratio decreases to 1 and almost achieves the theoretical value $P_{Ca^{2+}/Na^{+}}^{\lim}$ = 0.89. This effect is associated with the onset of limiting and overlimiting states on the CEM, and the membrane becomes nonselective. This effect is traditionally associated with the presence of the diffusion boundary layer, where decreasing transport of doubly charged ions is ascribed to its lower diffusivity in the solution. The authors of [48] investigated the Selemion CMV membrane permselectivity in the mixture of Na^+^/Mg^2+^ and discovered that increasing the current density (below the limiting condition) resulted in decreased transport number of Mg^2+^ and the permselectivity over Na^+^.

The limiting permeability coefficient in the mixed Ca^2+^/H^+^ solution equals 0.13, so the H^+^ flux should exceed Ca^2+^ one in 8 times. As can be seen from Figure 11b, Ca^2+^ and H^+^ fluxes have close values for the initial MK-40 membrane at low potentials per elementary cell (*U*_el.c._ < 1.5 V). At higher potentials permeability coefficient decreases to 0.6 but does not reach the theoretical value. After modification of the MK-40 membrane by the PANI, the dependence of Ca^2+^/H^+^ flux ratio rises on voltage per elementary cell. At low potentials, H^+^ flux through composite MK-40/PANI membrane is about two times higher than the Ca^2+^ one, which agrees with the estimation of counter-ion transport number within the extended three-wire model. At *U*_el.c._ > 3 V flux ratio becomes similar to the initial MK-40 membrane (Figure 11b). The authors of [49] have shown the predominant transport of Na^+^ ions through the membrane in the mixed Na^+^/Ca^2+^ solution under low current density. This effect is obtained for commercial monovalent cation-selective Astom CMS membrane and CMX membrane modified by quaternized poly-(2-vinylpyridine). The ion fluxes become nevertheless almost equal with increasing current density.

The presented data show that the PANI layer located on the surface of the heterogeneous membrane does not significantly affect the characteristics of the electrodialysis process. These results agree with the results of the investigation of physic-chemical characteristics of the membrane before and after modification. The expected decrease in doubly charged cations flux after modification by the PANI, however, is observed. The higher quantity of the PANI on the surface of CEM should enhance the permselectivity of the PANI modified CEMs.

#### 3.2.2. Homogeneous Membrane

The results of the electrodialysis treatment of the mixed solutions using homogeneous initial MF-4SK and composite MF-4SK/PANI membranes are presented in Figure 12 and Figure 13. One can see that modification of homogeneous membrane by the PANI also does not lead to an increase in energy consumption despite the substantial decrease in membrane conductivity after modification in all investigated solutions (Figure 4b). The energy consumption is generally comparable with the values for the electrodialysis with the heterogeneous membranes. The current efficiency for the electrodialysis treatment of NaCl + CaCl_2_ mixed solution using initial and modified membranes and HCl + CaCl_2_ solution using initial membrane are also close to the heterogeneous ones (Figure 12a–c). There are some peculiarities in the electrodialysis treatment of HCl + CaCl_2_ mixed solution with the MF-4SK/PANI membrane (Figure 12d): current efficiency for Ca^2+^-ions essentially decreases in comparison to H^+^, unlike heterogeneous membranes and salt solutions.

Figure 13 presents the theoretical and experimental flux ratios in Ca^2+^/Na^+^ and Ca^2+^/H^+^ solutions. As for the current efficiency curves, flux ratios in salt solution are similar for both types of membranes before and after modification. One can see that in Ca^2+^/H^+^ solution, the $J_{Ca^{2+}}^{}$ exceeds $J_{H^{+}}^{}$ for the MF-4SK membrane at *U*_el.c._ < 1.5 V, while the opposite relation is observed for the MK-40 membrane. This may be due to the relay-race mechanism of proton transfer, which should be provided by the presence of free water. Table 2 shows that the MF-4SK membrane has a lower volume fraction of free water than the MK-40 one, so it can be assumed that proton transport is restricted during the competitive transfer of Ca^2+^ and H^+^ ions. It is revealed the relation between the structure and properties of membranes and the efficiency of the separation of Ca^2+^/H^+^ ions during the electrodialysis. The most significant differences are observed for the MF-4SK/PANI membrane during the electrodialysis of acid solution (Figure 13b). A significant decrease in Ca^2+^ transfer in comparison with H^+^ and Na^+^ ions in the MF-4SK/PANI membrane is revealed. In contrast to the non-modified MF-4SK membrane, the double-/single-charged cations flux ratio achieves the limiting permeability value in a wide potential range for both Na^+^ and H^+^ counter-ions.

Such a result is obtained for the first time and has not been described in the literature. It is shown the applicability of composites on the basis of perfluorinated MF-4SK membranes and PANI for the separation of singly and doubly charged counter-ions by the electrodialysis. For the first time, it was shown that the permselectivity of the membrane could be observed not only in ohmic but in overlimiting operation regimes of the electrodialysis unit.

## 4. Conclusions

The present investigation has shown that the cation exchange membrane could be modified by polyaniline directly in the electrodialysis unit immediately before the electrodialysis separation of multicomponent solutions. The diffusion permeability, specific conductivity, current–voltage curves of the homogeneous and heterogeneous membranes in NaCl, HCl, CaCl_2_ solutions and their mixtures are investigated. The modification of the cation exchange membranes by polyaniline leads to the decrease of diffusion permeability and specific conductivity in all studies solutions. However, this effect is more significant for the modified homogeneous membrane than for the heterogeneous one. The diffusion permeability and specific conductivity of homogeneous MF-4SK membrane decrease after modification 2–3 times and 3–4 times, respectively. These characteristics for modified heterogeneous MK-40 membrane decrease 1.5 times and 1.4 times, correspondingly. The tree-wire model is used to calculate the counter-ion transport numbers for all samples. The decrease in the selectivity of the modified membranes in comparison with the initial ones is observed. The asymmetry of the current–voltage curves for modified homogeneous MF-4SK membrane is found, and the shape of the current–voltage curves depends on their orientation towards the counter-ion flow. Thus, the efficiency of modification of the cation exchange membrane by polyaniline substantially depends on the structural type of the membrane. In the case of a perfluorinated homogeneous membrane, the modification is more effective. The complex characterization of membranes can be used for the prediction of their permselectivity in the electrodialysis process.

The competitive mass transfer of singly and doubly charged cations in the electrodialysis process with the initial and modified cation exchanges membranes is investigated. The increase in membrane resistance after modification negligible influences the energy consumption of the electrodialysis process. The ratio of Ca^2+^/H^+^ fluxes through polyaniline-modified membrane equals 0.13, while for the initial membrane, this value varies in range 1.25–0.85. For the first time, it is shown that permselectivity of the membrane to singly charged cations in the mixture of singly/doubly charged cations could persist not only in underlimiting but also in overlimiting current regimes in potentials range 0.5–6 V per elementary cell. This result is obtained using polyaniline-modified homogeneous membrane in HCl/CaCl_2_ mixed solution.

## Figures and Tables

**Figure 1 membranes-11-00227-f001:**
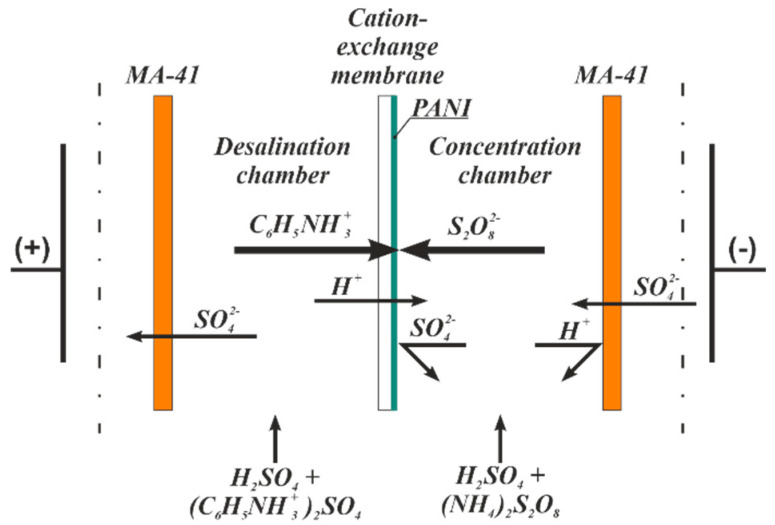
Synthesis of polyaniline (PANI) in the surface layer of the membrane in an elementary electrodialysis cell.

**Figure 2 membranes-11-00227-f002:**
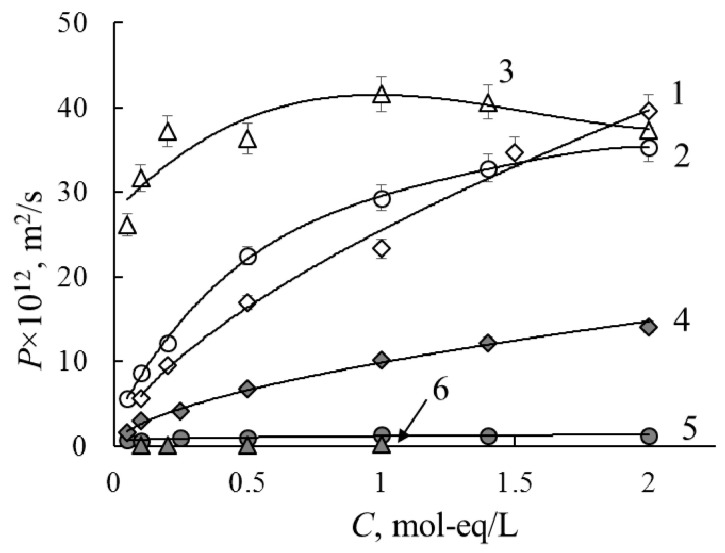
Diffusion permeability (DP) concentration dependencies for the MF-4SK (curves 1–3) and MF-4SK/PANI (curves 4–6) membranes in HCl (1, 4), NaCl (2, 5) and CaCl_2_ (3, 6) solutions.

**Figure 3 membranes-11-00227-f003:**
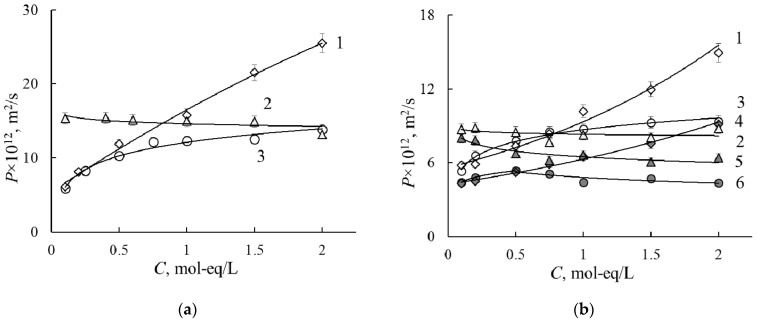
DP concentration dependencies for the MK-40 (**a**) and MK-40/PANI (**b**) membranes in HCl (1, 4), CaCl_2_ (2, 5) and NaCl (3, 6) solutions. Membrane oriented by the non-modified (white marks: 1, 2, 4) and modified (gray marks: 3, 5, 6) surfaces to electrolyte flux.

**Figure 4 membranes-11-00227-f004:**
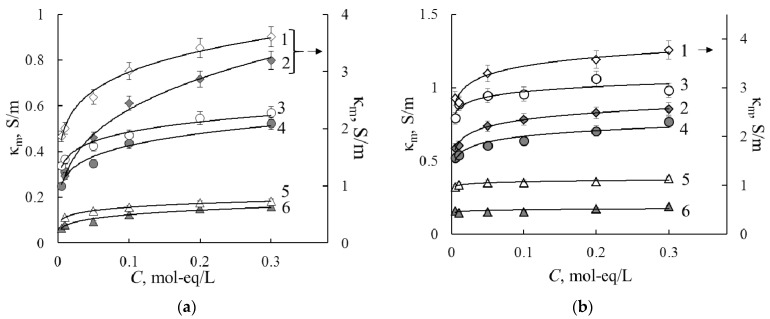
Conductivity concentration dependencies for the MK-40 (**a**), MK-40/PANI (**a**), MF-4SK (**b**) and MF-4SK/PANI (**b**) membranes in HCl (1, 2), NaCl (3, 4) and CaCl_2_ (5, 6) solutions.

**Figure 5 membranes-11-00227-f005:**
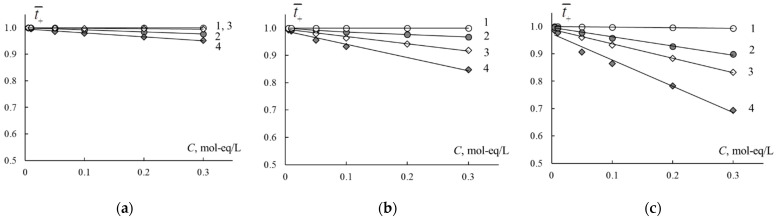
Transport numbers of counter-ions in the MF-4SK (1), MF-4SK/PANI (2), MK-40 (3) and MK-40/PANI (4) membranes in HCl (**a**), NaCl (**b**), CaCl_2_ (**c**) solutions.

**Figure 6 membranes-11-00227-f006:**
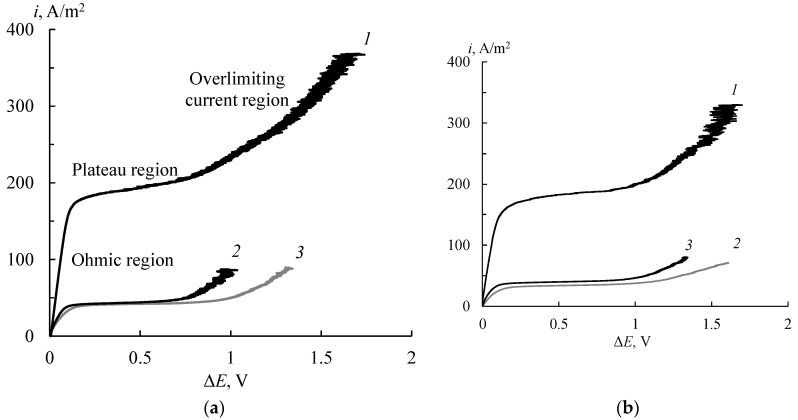
CVCs of the initial MF-4SK (**a**) and MK-40 (**b**) membranes in 0.05 mol-eq/L solutions of HCl (1), NaCl (2) and CaCl_2_ (3).

**Figure 7 membranes-11-00227-f007:**
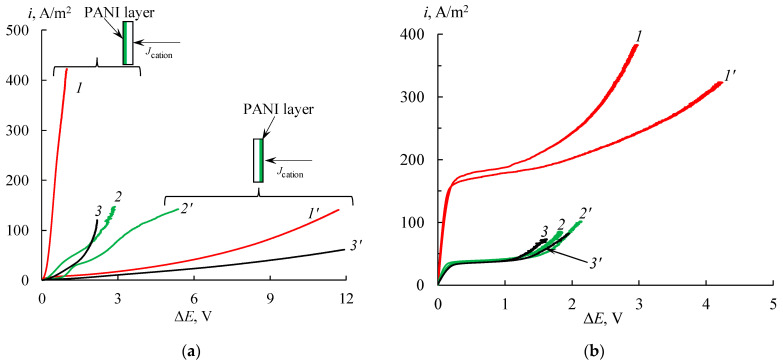
CVCs of modified MF-4SK/PANI (**a**) and MK-40/PANI (**b**) in 0.05 mol-eq/L HCl (1, 1′), NaCl (2, 2′) and CaCl_2_ (3, 3′). The composite is faced with the non-modified surface (1, 2, 3) and PANI layer (1′, 2′, 3′) to the counter-ions flow.

**Figure 8 membranes-11-00227-f008:**
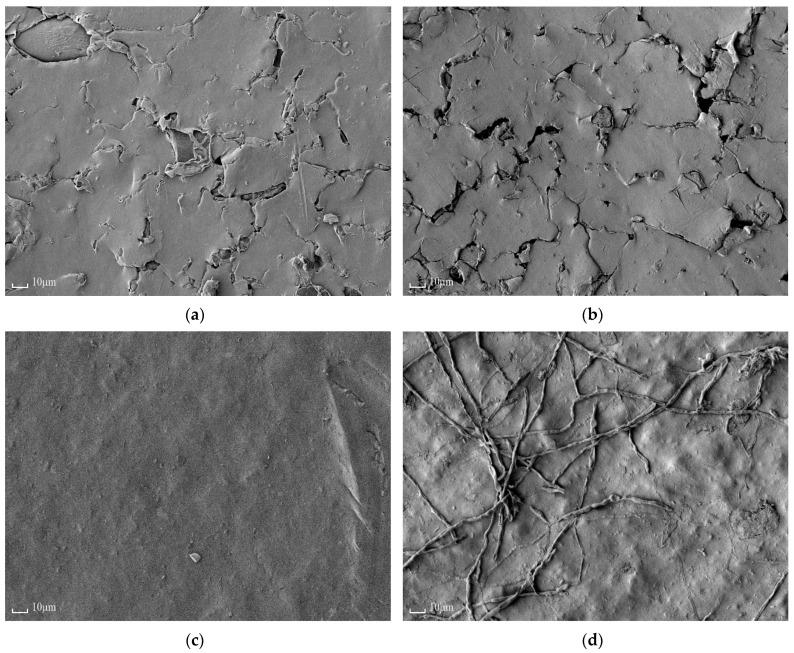
SEM images of the non-modified (**a**,**c**) and modified (**b**,**d**) surfaces of the MK-40/PANI (**a**,**b**) and MF-4SK/PANI (**c**,**d**) membranes.

**Figure 9 membranes-11-00227-f009:**
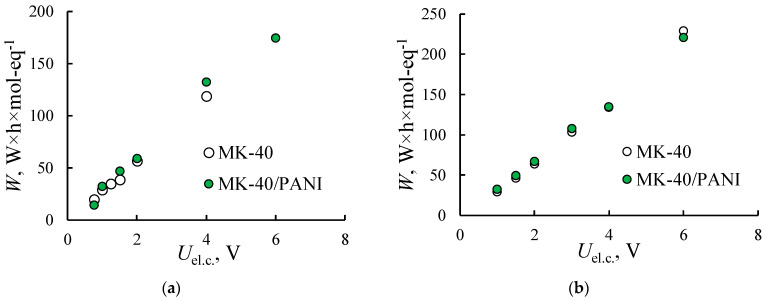
Energy consumption for the electrodialysis treatment of NaCl + CaCl_2_ (**a**) and HCl + CaCl_2_ (**b**) solutions with the initial MK-40 and composite MK-40/PANI membranes.

**Figure 10 membranes-11-00227-f010:**
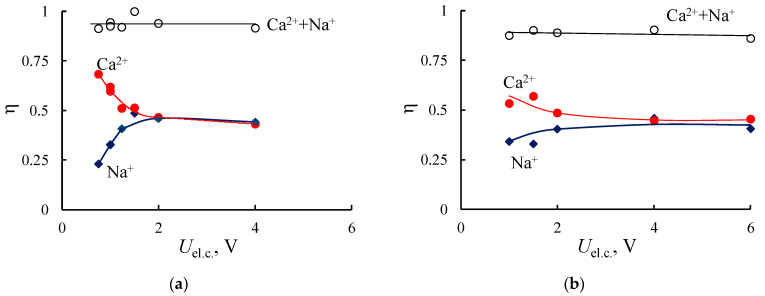
Current efficiency for the electrodialysis processing of NaCl + CaCl_2_ (**a**,**b**) and HCl + CaCl_2_ (**c**,**d**) solutions with the initial MK-40 (**a**,**c**) and composite MK-40/PANI (**b**,**d**) membranes.

**Figure 11 membranes-11-00227-f011:**
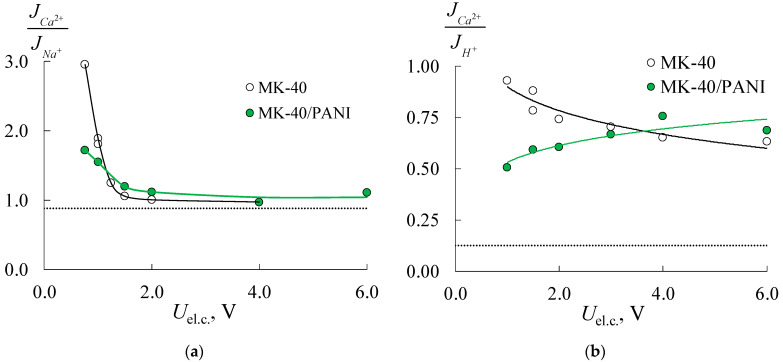
Theoretical (dashed line) and experimental (points and solid lines) dependencies of Ca^2+^/Na^+^ (**a**) and Ca^2+^/H^+^ (**b**) flux ratios for the initial MK-40 and composite MK-40/PANI membranes.

**Figure 12 membranes-11-00227-f012:**
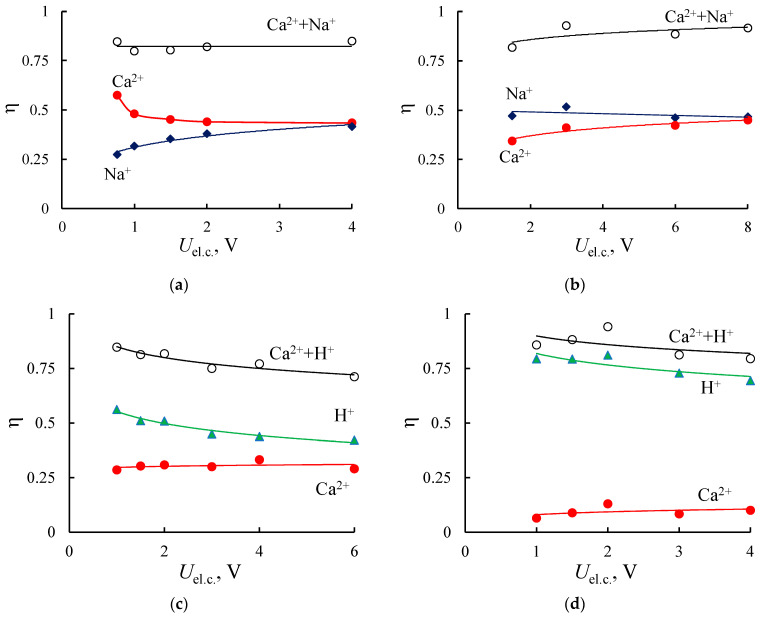
Current efficiency for the elecrodialysis processing of NaCl + CaCl_2_ (**a**,**b**) and HCl + CaCl_2_ (**c**,**d**) solutions with the initial MF-4SK (**a**,**c**) and composite MF-4SK/PANI (**b**,**d**) membranes.

**Figure 13 membranes-11-00227-f013:**
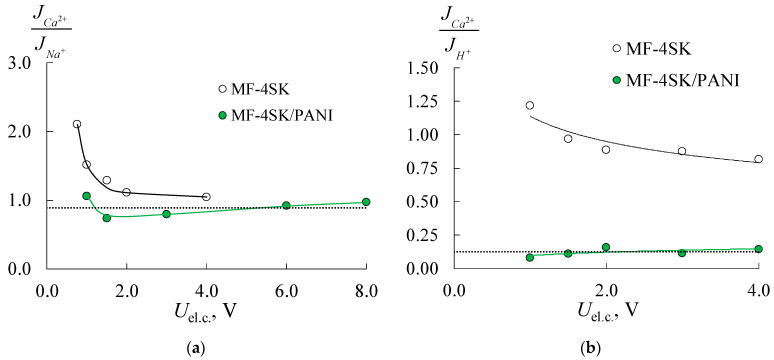
Theoretical (dashed line) and experimental (points and solid lines) dependencies of Ca^2+^/Na^+^ (**a**) and Ca^2+^/H^+^ (**b**) flux ratios for the initial MF-4SK and composite MF-4SK/PANI membranes.

**Table 1 membranes-11-00227-t001:** The physical-chemical properties of the initial and modified membranes.

Membrane	IEC, mmol/g_sw_	Water Content, %	Density, g_sw_/cm^3^	Specific Water Content, mol H_2_O/ mol SO_3_^−^
MK-40	1.54 ± 0.04	37	1.174	13
MK-40/PANI	1.55 ± 0.04	40	1.12	14
MF-4SK	0.68 ± 0.04	20	1.70	16
MF-4SK/PANI	0.70 ± 0.04	21	1.68	17

**Table 2 membranes-11-00227-t002:** Transport-structural parameters of the extended tree-wire model.

Membrane	Electrolyte	*f* _2_	α	*a*	*b*	*c*	*d*
MK-40	HCl	0.18	0.38	0.40	0.59	0.011	0.43
MK-40/PANI		0.22	0.59	0.27	0.65	0.078	0.54
MF-4SK		0.08	0.31	0.24	0.76	2.9 × 10^−4^	0.34
MF-4SK/PANI		0.12	0.46	0.24	0.75	0.0102	0.47
MK-40	NaCl	0.15	0.53	0.23	0.74	0.026	0.52
MK-40/PANI		0.21	0.51	0.32	0.64	0.045	0.50
MF-4SK		0.05	0.40	0.12	0.88	6.0 × 10^−4^	0.42
MF-4SK/PANI		0.12	0.51	0.20	0.79	0.014	0.51
MK-40	CaCl_2_	0.19	0.41	0.38	0.61	0.016	0.45
MK-40/PANI		0.25	0.38	0.51	0.47	0.026	0.44
MF-4SK		0.03	0.53	0.051	0.95	12 × 10^−4^	0.53
MF-4SK/PANI		0.14	0.43	0.29	0.70	0.010	0.45

## Data Availability

The data presented in this study are available on request from the corresponding author.

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
