# Peer review of "Permselectivity of Cation Exchange Membranes Modified by Polyaniline"

_membranes, 2021, doi:10.3390/membranes11030227_

Round 1

Reviewer 1 Report

- The abstract and conclusion parts are described qualitatively, and more quantitative debate and its conclusive remark should be suggested for improvement of the present manuscript.

- There are lacking experimental sections that can explain how to measure the values discussed in Table 1.

- Materials in the present study are well discussed for comparison, but a comparison to the state-of-the-art materials in this field is not available in the present manuscript.

- Many equations are described at several sections of the manuscript, and they seem to be very general, not so special in the present study. Thus, I recommend the authors to refer to corresponding references to simplify the manuscript.

- I strongly recommend the authors to carefully review the English grammars used in the manuscript. Sometimes, sentences do not consist of adequate ways, such as S, V, O, C.

Reviewer 2 Report

The report is in the enclosed file.

Round 2

Reviewer 1 Report

The revised manuscript has been improved to a certain extent, in order to suffice all requests of the former points, thus I recommend to publish it as it is.